# Improved clinical data imputation via classical and quantum determinantal point processes

**Skander Kazdaghli[1]\*, Iordanis Kerenidis[1,2], Jens Kieckbusch[3], Philip Teare[4]**

[1]QC Ware, Paris, France; [2]Universite de Paris, CNRS, IRIF, Paris, France; [3]Emerging Innovations Unit, Discovery Sciences, BioPharmaceuticals R&D, AstraZeneca, Cambridge, United Kingdom; [4]Centre for AI, Data Science & AI, BioPharmaceuticals R&D, AstraZeneca, Cambridge, United Kingdom

**Abstract** Imputing data is a critical issue for machine learning practitioners, including in the life sciences domain, where missing clinical data is a typical situation and the reliability of the imputation is of great importance. Currently, there is no canonical approach for imputation of clinical data and widely used algorithms introduce variance in the downstream classification. Here we propose novel imputation methods based on determinantal point processes (DPP) that enhance popular techniques such as the multivariate imputation by chained equations and MissForest. Their advantages are twofold: improving the quality of the imputed data demonstrated by increased accuracy of the downstream classification and providing deterministic and reliable imputations that remove the variance from the classification results. We experimentally demonstrate the advantages of our methods by performing extensive imputations on synthetic and real clinical data. We also perform quantum hardware experiments by applying the quantum circuits for DPP sampling since such quantum algorithms provide a computational advantage with respect to classical ones. We demonstrate competitive results with up to 10 qubits for small-scale imputation tasks on a state-of-the-art IBM quantum processor. Our classical and quantum methods improve the effectiveness and robustness of clinical data prediction modeling by providing better and more reliable data imputations. These improvements can add significant value in settings demanding high precision, such as in pharmaceutical drug trials where our approach can provide higher confidence in the predictions made.

## eLife assessment

The methods presented in this work provide modest yet consistent accuracy improvements for data classification tasks where certain data are missing. The authors also present a way to use quantum computers for this task. The methodology and results for the classical (non-quantum) case are **solid**, although evidence for the practical quantum advantage via their approach in 'next generation' quantum computers remains **incomplete**. The results are **valuable** and should interest data scientists, life scientists and anyone working in quantum computing.

## Introduction

Missing data is a recurring problem in machine learning and in particular for clinical datasets, where it is common that numerous feature values are not present for reasons including incomplete data collection and discrepancies in data formats and data corruption (*Luo, 2022*; *Emmanuel et al., 2021*; *Pedersen et al., 2017*; *Myers, 2000*). Machine learning is routinely used in life science and clinical research for prediction tasks, such as diagnostics (*Qin et al., 2019*) and prognostics (*Booth*

\*For correspondence:
skander.kazdaghli@gmail.com

**Figure 1.** Example of overall workflow for patient management through clinical data imputation and downstream classification.

_et al., 2021_), as well as estimation tasks, such as biomarker proxies (_Wang et al., 2017_) and digital biomarkers (_Rendleman et al., 2019_). Beyond the research setting, machine learning is becoming more commonplace as regulated Software as a Medical Device, where machine learning models are influencing – or making – clinical decisions that affect patient care.

Machine learning algorithms typically require complete datasets and missing values can significantly affect the quality of the machine learning models trained on such data. This is in large part due to the fact that there can be different underlying reasons for the missingness: for example, feature values can be missing completely at random (MCAR), missing at random (MAR), and missing not at random (MNAR), each one with their own characteristics.

Despite its importance for clinical trials, there is no canonical approach for dealing with missingness and finding appropriate, effective and reproducible methods remains a challenge. A common way to deal with missing clinical data is to exclude subjects that do not have the complete set of feature values present. A drawback of this approach is that excluding subjects can in fact introduce significant biases in the final model. For example, it can result in the model being trained to be more effective for the type of subjects that are likely to have complete data than for those that do not. Moreover, the effectiveness and reliability of clinical trials are reduced when subjects with missing feature values are excluded from the clinical trial.

Data imputation is an alternative to the complete dataset approach, where subjects with missing feature values are not excluded. Instead, missing values are imputed to create a complete dataset that is then used for a classification task as shown in _Figure 1_. There are different ways to achieve this, including 'filling' missing values with zeros, or with the mean value of the feature across all subjects that have such a value present. These methods provide consistent imputation results, but there are important caveats for using such simple methods since they ignore possible correlations between features and can make the dataset appear more homogeneous than it really is. More advanced data imputation methods have been proposed in the literature: iterative methods include the multivariate imputation by chained equations (MICE) (_Groothuis-Oudshoorn, 2011_) and MissForest (_Stekhoven and Bühlmann, 2012_) algorithms, and deep learning methods include GAIN (generative adversarial imputation nets) _Yoon and Jordon, 2018_ and MIWAE (missing data importance-weighted autoencoder) (_Mattei and Frellsen, 2019_). Recent results _Shadbahr et al., 2022_ have shown that for clinical data two iterative imputation methods, MiceRanger, which uses predictive mean matching, and MissForest, which uses Random Forests to predict the missing values of each feature using the other features, provide the best results and have been used here as a baseline.

Several metrics are routinely used to quantify the quality of data imputation: point-wise discrepancy measures include root mean square error, mean absolute error, and coefficient of determination ($R^2$). Feature-wise discrepancy measures include Kullback–Leibler divergence, two-sample Kolmogorov–Smirnov statistic or (2-)Wasserstein distance. Ultimately, the quality and reliability of imputations can be measured by the performance of a downstream predictor, which is usually the area under the receiver operating curve (AUC) for a classification task. In practical terms, the performance of the

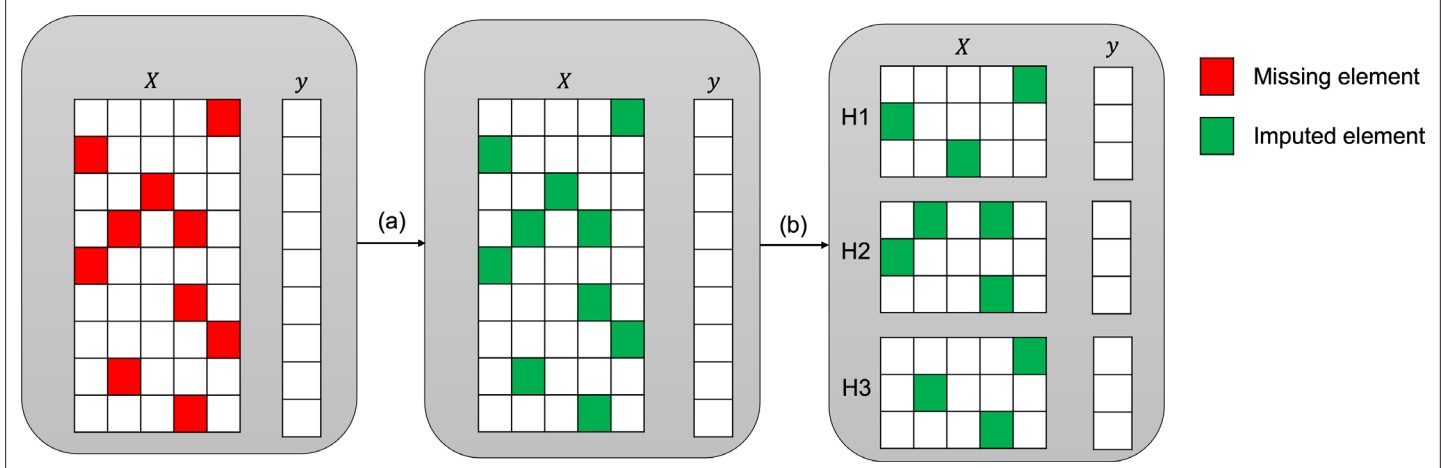

**Figure 2.** Imputation and downstream classification procedure to benchmark the imputation method's performance. First, the imputer is trained on the whole observed dataset X as shown in step (a). In step (b), the imputed data is split into three consecutive folds (holdout sets H1, H2, and H3), then a classifier is trained on each combination of two holdout sets (development sets D1, D2, and D3) and the area under the receiver operating curve (AUC) is calculated for each holdout set.

downstream classifier is usually of highest importance for clinical datasets: for example, in one of our datasets, the classifier denominates a binary outcome of a critical care unit stay (e.g., survival) for each patient. Accordingly, we have used AUC for the classification task here on different holdout sets (see *Figure 2*) to assess the performance of our novel methods.

In order to increase the resulting AUC, we combine the MissForest and MiceRanger imputation methods with determinantal sampling, based on determinantal point processes (DPP) (*Dereziński and Mahoney, 2021*; *Kulezsa and Taskar, 2011*), which favors samples that are diverse and thus reduces the variance of the training of each decision tree that in turn provides more accurate models. In essence, determinantal sampling picks subsets of data according to a distribution that gives more weight to subsets of data that contain diverse data points. More precisely, each subset of data points is picked according to the volume encapsulated by these data points. The determinantal distribution increases the attention given to uncommon or out-of-the-ordinary data points rather than biasing the learning process towards the more commonly found data, which can improve the overall prediction accuracy in particular for unbalanced datasets as is often the case for clinical data (Dereziński and Mahoney). Determinantal sampling for regression and classification tasks with full data has been proposed previously for linear regressors (*Dereziński et al., 2018*) and for Random Forest training for a financial data classification use case where it outperformed the standard Random Forest model (*Thakkar et al., 2023*). However, an inherent feature of standard Random Forest and determinantal sampling algorithms is randomness that produces data imputations that vary from one run of the algorithm to the next. This is often undesirable since the downstream classification performance can also be affected, which motivated us to apply a deterministic version of determinantal sampling (*Schreurs and Suykens, 2021*) within the Random Forests of the imputation methods to provide more robust and reliable imputations.

Through deterministic determinantal sampling, we address two challenges in data imputation: first, we provide improved data imputation methods that can increase the performance of the downstream classifier; and second, we remove the variance of the common stochastic and multiple imputation methods, thus ensuring reproducibility, easier integration in machine learning workflows, and compliance with healthcare regulations. While these improvements are of particular relevance for clinical data, our algorithms can also be advantageous for other imputation tasks where improving downstream classification and removing variance is of importance.

In order to demonstrate this improvement, we apply our methods to two classification datasets: a synthetic dataset and a public clinical dataset where the predicted outcome is the survival of the patient.

In addition, we explore the potential of quantum computing to speed up these novel imputation methods: we provide a quantum circuit implementation of the determinantal sampling algorithm

that offers a computational advantage compared to its classical counterpart. The best classical algorithms for determinantal sampling take in practice cubic time in the number of features to provide a sample (*Dereziński and Mahoney, 2021*). In contrast, the quantum algorithm we present here, based on theoretical analysis in *Kerenidis and Prakash, 2022*, has running time that scales linearly with the number of features. We measure running time as the depth of the necessary quantum circuits, given that the quantum processing units that are being developed currently offer the possibility of performing parallel operations on disjoint qubits.

This suggests that with the advent of next-generation quantum computers with more and better qubits, one could also expect a computational speedup in performing determinantal sampling using a quantum computer. Here, we demonstrate competitive results with up to 10 qubits for small-scale imputation tasks on a state-of-the-art IBM quantum processor.

This work combines classical (*Dereziński and Mahoney, 2021*) and quantum (*Kerenidis and Prakash, 2022*) DPP algorithms with widely used data imputation methods, resulting in novel data imputation algorithms that can improve performance on classical computers while also having the potential of a quantum speedup in the future.

## Results

In 'Methods', we provide a detailed description of our four imputation methods, DPP-MICE, DPP-MissForest, detDPP-MICE, and detDPP-MissForest. All of them are based on iterative imputation methods that use the observed values of every column to predict the missing values. The model used to fill missing values in each column is the Random Forest classifier. Our imputation methods replace the standard Random Forest used by the original miceRanger and MissForest imputers by the DPP-Random Forest model, for our first two imputers, and the detDPP-Random Forest for the latter two. The DPP-Random Forest model subsamples the data for each decision tree using determinantal sampling instead of uniform sampling, while the detDPP-Random Forest model deterministically picks for each decision tree the subset of data that has the maximum probability according to the determinantal distribution. We also demonstrate a computationally advantageous way to perform the determinantal sampling on quantum computers.

In order to benchmark the different imputation methods, we used two types of datasets with a categorical outcome variable. First, a synthetic dataset, created using the scikit-learn method *make_classification*. It consists of 2000 rows with 25 informative features. This is useful to study the imputation quality where features have equal importance. Second, the MIMIC-III dataset (*Johnson et al., 2016*): the Medical Information Mart for Intensive Care (MIMIC) dataset, which is a freely available clinical database. It is comprised of data for patients who stayed in critical care units at the Beth Israel Deaconess Medical Center between 2001 and 2012. It contains the data of 7214 patients with 14 features.

We also applied two types of missingness on these datasets: MCAR, where the missingness distribution is independent of any observed or unobserved variable, and MNAR, where the missingness distribution depends on the outcome variable. We expect similar results to hold for the MAR case as well, but it was not considered in this work.

We present the numerical results in terms of the AUC of the downstream classification task in *Table 1* and provide graphs of the results in *Tables 2 and 3*. Each experiment was run 10 times with different random seeds to get the variance of the results.

Overall, DPP-MICE and DPP-MissForest provide improved results compared to their classical baseline MICE and MissForest. This is the case for both the synthetic and the MIMIC datasets and for both MCAR and MNAR missingness. Even more interestingly, the detDPP-MICE and detDPP-MissForest collapse the variance of the imputed data to 0 and moreover lead in most cases to even higher AUC than the expectation of the previous methods.

### DPP-MICE and detDPP-MICE outperform MICE

We present the performance results of MICE-based methods in terms of the AUC of the downstream classification task using an XGBoost classifier, which has been shown to be the strongest classifier for such datasets (*Shadbahr et al., 2022*). We used the default parameters of the classifier since our focus is comparing the different imputation methods. In each case, the original dataset with induced

**Table 1.** AUC results for the SYNTH and MIMIC-III datasets, with MCAR and MNAR missingness, three holdout sets, and six different imputation methods.

Values are expressed as mean ± SD of 10 values for each experiment. DPP-MICE and detDPP-MICE are in bold when outperforming MICE and the underlined one is the best of the three. DPP-MissForest and detDPP-MissForest are in bold when outperforming MissForest and the underlined one is the best of the three.

| Dataset | Missingness | Set | MICE | DPP-MICE | detDPP-MICE | MissForest | DPP-MissForest | detDPP-MissForest |
|---|---|---|---|---|---|---|---|---|
| | | H1 | 0.8318 ± 0.0113 | 0.835 ± 0.0083 | 0.8352 | 0.8525 ± 0.0044 | 0.8552 ± 0.0049 | 0.8582 |
| | | H2 | 0.8316 ± 0.008 | 0.8369 ± 0.0128 | 0.84 | 0.8465 ± 0.0057 | 0.849 ± 0.003 | 0.8491 |
| | MCAR | H3 | 0.8205 ± 0.0127 | 0.8266 ± 0.0096 | 0.8272 | 0.8436 ± 0.0031 | 0.8452 ± 0.0048 | 0.855 |
| | | H1 | 0.8903 ± 0.0046 | 0.8915 ± 0.007 | 0.8934 | 0.7133 ± 0.0063 | 0.7171 ± 0.01 | 0.7185 |
| | | H2 | 0.8755 ± 0.01 | 0.8745 ± 0.0072 | 0.8955 | 0.7052 ± 0.0036 | 0.7124 ± 0.0078 | 0.7167 |
| SYNTH | MNAR | H3 | 0.9003 ± 0.0059 | 0.9005 ± 0.006 | 0.9041 | 0.769 ± 0.0103 | 0.7773 ± 0.0129 | 0.7905 |
| | | H1 | 0.7621 ± 0.0046 | 0.7628 ± 0.0049 | 0.7641 | 0.7687 ± 0.0012 | 0.77 ± 0.0013 | 0.771 |
| | | H2 | 0.7541 ± 0.0037 | 0.7532 ± 0.0047 | 0.7619 | 0.7649 ± 0.0019 | 0.777 ± 0.0019 | 0.7707 |
| | MCAR | H3 | 0.7365 ± 0.0055 | 0.7394 ± 0.0052 | 0.7471 | 0.7485 ± 0.001 | 0.7507 ± 0.0017 | 0.7515 |
| | | H1 | 0.77 ± 0.0026 | 0.7717 ± 0.0036 | 0.7722 | 0.6616 ± 0.0065 | 0.6715 ± 0.07 | 0.6760 |
| | | H2 | 0.777 ± 0.0064 | 0.7818 ± 0.0029 | 0.7812 | 0.6748 ± 0.0045 | 0.6778 ± 0.0048 | 0.6798 |
| MIMIC | MNAR | H3 | 0.7324 ± 0.0047 | 0.7363 ± 0.0031 | 0.7403 | 0.6368 ± 0.0034 | 0.64 ± 0.004 | 0.6419 |

AUC = area under the receiver operating curve; MCAR = missing completely at random; MNAR = missing not at random.

missing values is imputed using MICE, DPP-MICE, or detDPP-MICE, then it is divided into threefolds of development/holdout sets. The downstream classifier is then trained on each development set and its performance is measured by the AUC for the corresponding holdout set. The results are shown in *Table 1* and in the figures in *Table 2*.

The imputation procedure is performed for a total of 10 iterations over all the columns and for each column, a (DPP) Random Forest regressor is trained using 10 trees. For each Random Forest training, the dataset is divided into batches of 150 points each and DPPs are used to sample from every batch.

The results show that across the 12 in total dataset experiments DPP-MICE outperforms MICE on expectation in 10 of them, while detDPP-MICE provides a single deterministic imputation, which outperforms the expected result from MICE in all 12 datasets and from DPP-MICE 11 out of 12 times.

## DPP-MissForest and detDPP-MissForest outperform MissForest

Here we present the performance results of MissForest-based methods in terms of the AUC of the downstream classification task using again an XGBoost classifier. In each case, the original dataset with induced missing values is imputed using MissForest, DPP-MissForest, or detDPP-MissForest, then it is divided into threefolds of development/holdout sets. The downstream classifier is again then trained on each development set and its performance is measured by the AUC for the corresponding

**Table 2.** AUC results on the different holdout sets after imputation using MICE, DPP-MICE, and detDPP-MICE. In the case of MICE and DPP-MICE, the boxplots correspond to 10 AUC values for 10 iterations of the same imputation and classification algorithms, depicting the lower and upper quartiles as well as the median of these 10 values. The AUC values are the same for every iteration of the detDPP-MICE algorithm.

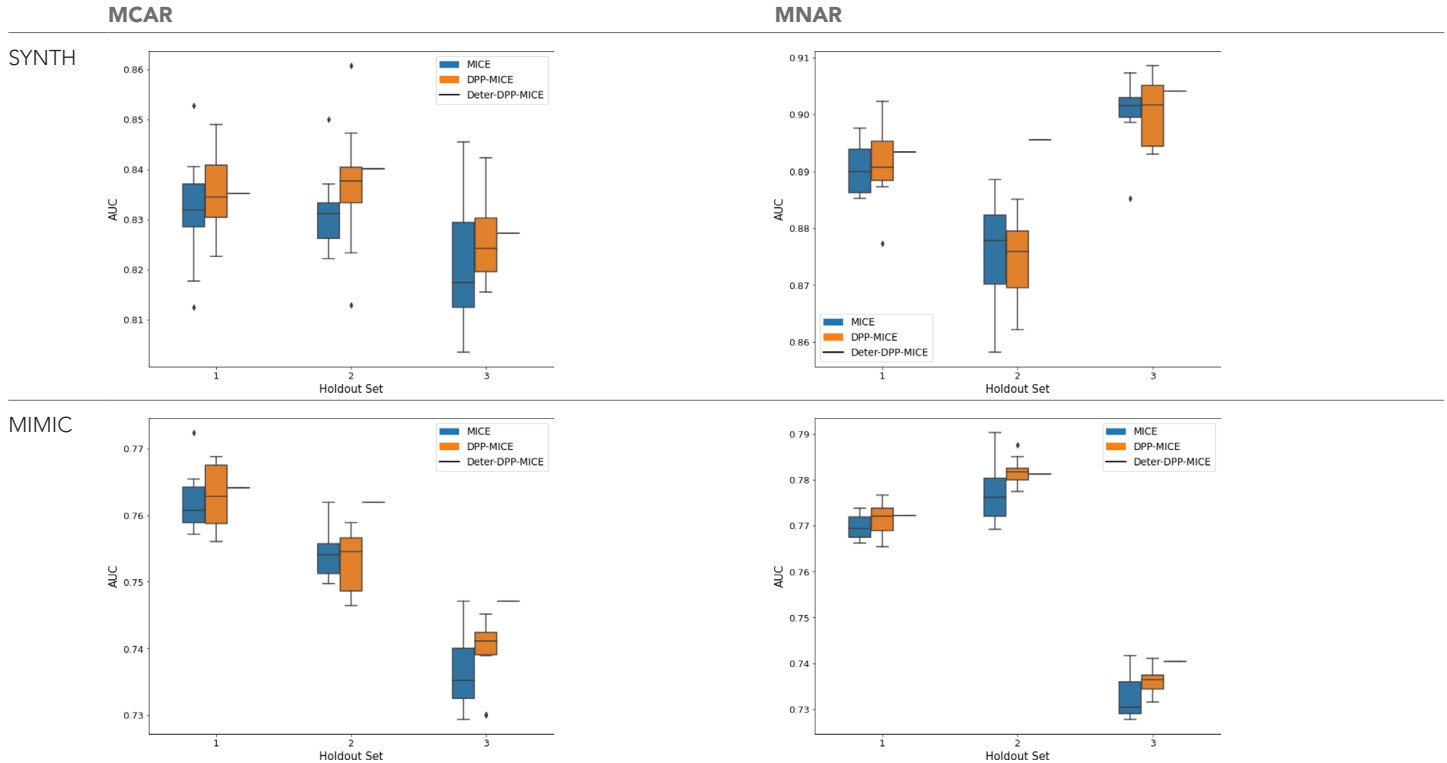

AUC = area under the receiver operating curve; MCAR = missing completely at random; MNAR = missing not at random.

holdout set. The results are shown in *Table 1* and in the figures in *Table 3*. The specifics of the Random Forest training are the same as in the case of MICE.

The results show that across all experiments DPP-MissForest outperforms MissForest in all 12 of them, while detDPP-MissForest provides a single deterministic imputation that outperforms the expected result from MissForest in all 12 datasets and from DPP-MissForest in 11 out of 12 times.

## Quantum hardware implementation of DPP-MissForest results in competitive downstream classification

As we describe in 'Methods', quantum computers can in principle be used to offer a computational advantage in determinantal sampling. In order to better understand the state of the art of current quantum hardware, we used a currently available quantum computer to perform determinantal sampling within a DPP-MissForest imputation method for scaled-down versions of the synthetic and MIMIC datasets.

- Reduced synthetic dataset: 100 points and three features, created using the sklearn method *make_classification*.
- Reduced MIMIC dataset: 200 points and three features. The three features were chosen from the original dataset features based on low degree of missingness and their predictiveness of the downstream classifier, and they were 'Oxygen saturation std', 'Oxygen saturation mean', and 'Diastolic blood pressure mean'.

For the purposes of our experiments, we used the 'ibm_hanoi' 27-qubit quantum processor shown in *Figure 3*. We implemented quantum circuits with up to 10 qubits. We also performed quantum simulations using the qiskit noiseless simulator. The decision trees of the DPP-Random Forests used by the imputation models are trained using batches of decreasing sizes (see *Table 4*). For example,

**Table 3.** AUC results on the different holdout sets after imputation using MissForest, DPP-MissForest, and detDPP-MissForest. In the case of MissForest and DPP-MissForest, the boxplots correspond to 10 AUC values for 10 iterations of the same imputation and classification algorithm, depicting the lower and upper quartiles as well as the median of these 10 values. The AUC values are always the same for every iteration of the detDPP-MissForest algorithm.

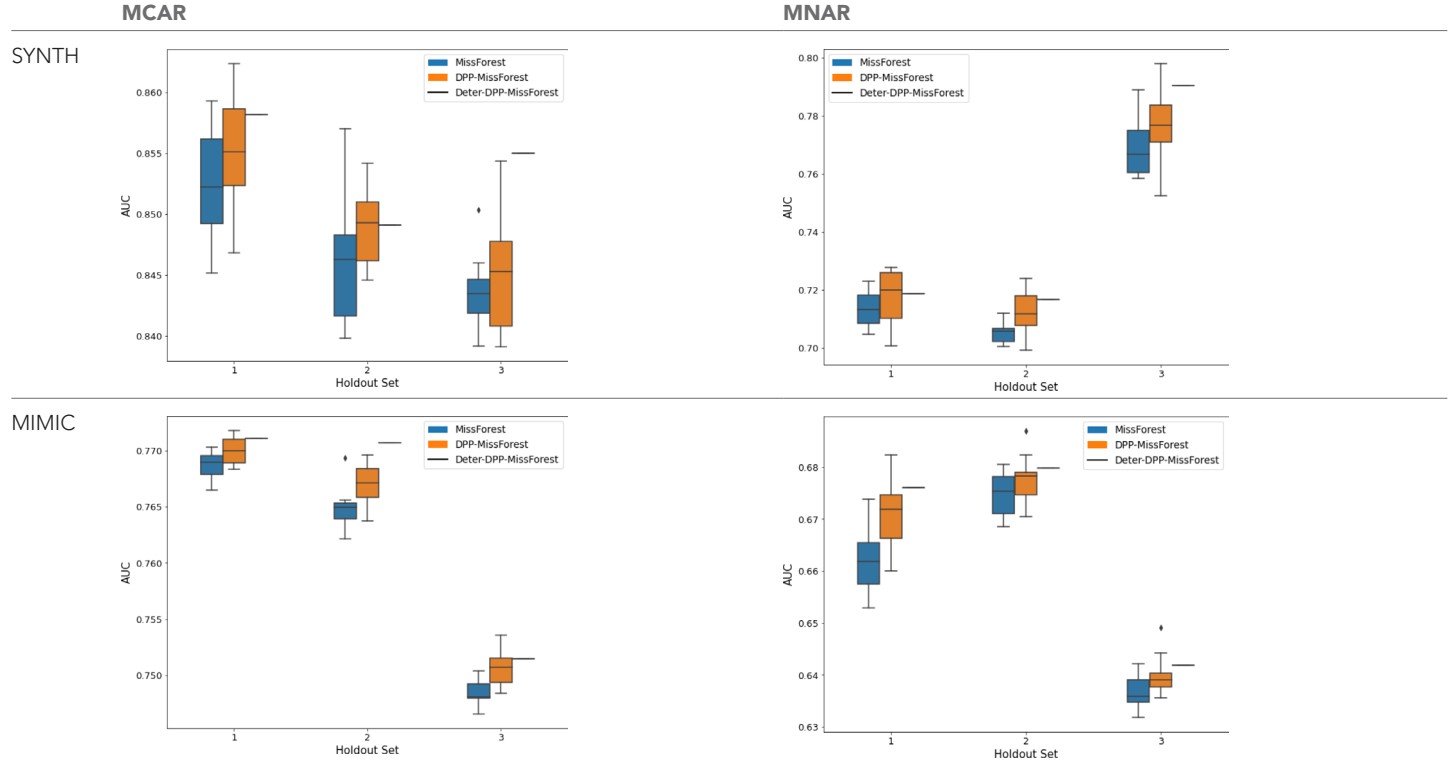

AUC = area under the receiver operating curve; MCAR = missing completely at random; MNAR = missing not at random.

for the algorithm with batch size equal to 10, the algorithm first samples 2 out of the 10 data points to use for the first decision tree, then from the remaining 8 data points it picks another 2 for the second tree, then 2 from the remaining 6, and last 2 from the remaining 4. In other words, we train four different trees, and each time we use a quantum circuit with number of qubits equal to 10, 8, 6, and 4, to perform the respective determinantal sampling.

In the figures of *Table 5* and in *Table 6*, we provide for the different dataset experiments the AUC for MissForest, the simulated results of the quantum version of DPP-MissForest, and the actual hardware experimental results of running the quantum version of DPP-MissForest. Even for these very small datasets, when simulating the quantum version of DPP-MissForest, we demonstrate an increase in the AUC compared to the MissForest algorithm. This further highlights the potential advantages of determinantal sampling within imputation methods. Of note, running our algorithms on current hardware introduces variance in the downstream classifier. Importantly, this variance is due to noise in the hardware rather than inherent to the algorithm.

Our quantum hardware results are competitive with standard methods and in many cases close to the values expected from the simulation. In some cases, we observed a clear deterioration of the AUC due to the noise and errors in the quantum hardware. The results are closer to the simulations when using MCAR missingness with larger batch sizes that use more trees both for synthetic and the MIMIC datasets. As explained above, even though the algorithm with batch size 10 means

**Table 4.** Data matrix sizes used by the quantum determinantal point processes (DPP) circuits to train each tree.

The number of rows corresponds to the number of data points and is equal to the number of qubits of every circuit.

| Batch size | Tree 1 | Tree 2 | Tree 3 | Tree 4 |
|---|---|---|---|---|
| 7 | (7,2) | (5,2) | - | - |
| 8 | (8,2) | (6,2) | (4,2) | - |
| 10 | (10,2) | (8,2) | (6,2) | (4,2) |

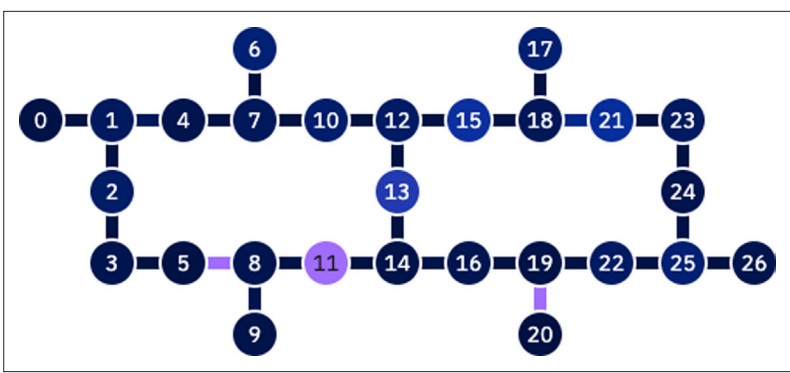

**Figure 3.** IBM Hanoi 27-qubit quantum processor.

**Table 5.** Hardware results using the IBM quantum processor, depicting AUC results of the downstream classifier task after imputing missing values using DPP-MissForest.

In the case of MissForest and the quantum hardware DPP-MissForest implementations, the boxplots correspond to 10 AUC values for 10 iterations of the same imputation and classification algorithm, depicting the lower and upper quartiles as well as the median of these 10 values. The AUC values are the same for every iteration of the quantum DPP-MissForest algorithm using the simulator.

|  | Batch size: 7 Number of trees: 2 | Batch size: 8 Number of trees: 3 | Batch size: 10 Number of trees: 4 |
|---|---|---|---|
| MCAR SYNTH | | | |
| MCAR MIMIC | | | |
| MNAR SYNTH | | | |
| MNAR MIMIC | | | |

AUC = area under the receiver operating curve; MCAR = missing completely at random; MNAR = missing not at random.

**Table 6.** Numerical quantum hardware results showing the AUC results of the downstream classifier task on reduced datasets.

Values are represented according to mean ± SD format given 10 values for each experiment.

| Dataset | Missingness | Batch size | Trees | MissForest | detDPP-MissForest (simulator) | detDPP-MissForest (hardware) |
|---------|-------------|-----------|-------|------------|-------------------------------|------------------------------|
|         |             | 7 | 2 | 0.868 ± 0.0302 | 0.9026 | 0.8598 ± 0.021 |
|         |             | 8 | 3 | 0.8667 ± 0.0342 | 0.9256 | 0.8923 ± 0.027 |
|         | MCAR        | 10 | 4 | 0.8725 ± 0.0275 | 0.9028 | 0.8902 ± 0.024 |
|         |             | 7 | 2 | 0.7122 ± 0.0264 | 0.78 | 0.7149 ± 0.02 |
|         |             | 8 | 3 | 0.7153 ± 0.022 | 0.729 | 0.7036 ± 0.0167 |
| SYNTH   | MNAR        | 10 | 4 | 0.7258 ± 0.0157 | 0.7868 | 0.7082 ± 0.036 |
|         |             | 7 | 2 | 0.7127 ± 0.038 | 0.7522 | 0.7117 ± 0.0315 |
|         |             | 8 | 3 | 0.7136 ± 0.03 | 0.7728 | 0.7448 ± 0.0258 |
|         | MCAR        | 10 | 4 | 0.6968 ± 0.03 | 0.7327 | 0.7262 ± 0.0299 |
|         |             | 7 | 2 | 0.7697 ± 0.0133 | 0.7794 | 0.7742 ± 0.0108 |
|         |             | 8 | 3 | 0.7713 ± 0.0112 | 0.7943 | 0.767 ± 0.0125 |
| MIMIC   | MNAR        | 10 | 4 | 0.7712 ± 0.0116 | 0.7922 | 0.7675 ± .01545 |

AUC = area under the receiver operating curve; MCAR = missing completely at random; MNAR = missing not at random.

using a quantum circuit with 10 qubits, the fact that we use four trees overall with a decreasing number of data points each time, and thus a decreasing number of qubits (namely, 10, 8, 6, and 4), results in an overall more reliable imputation.

## Discussion

Missing data is a critical issue for machine learning practitioners as complete datasets are usually required for training machine learning algorithms. To achieve complete datasets, missing values are usually imputed. In the case of clinical data, missing values and imputation can be a potential source of bias and can considerably influence the robustness and interpretability of results. Nevertheless, there is no canonical way to deal with missing data, which makes improvements in data imputation methods an attractive and impactful approach to increase the effectiveness and reliability of clinical trials. In this proof-of-concept study, we assessed the downstream consequences of implementing such improvements focussing on MCAR and MNAR to assess the usefulness of our approach. MNAR and MCAR represent two extreme cases of missingness with importance for clinical data imputation applications.

Determinantal point processing methods increase the diversity of the data picked to train the models, showcasing also that data gathering and preprocessing are important to remove biases related to over-representation of particular data types. This is more important when dealing with unbalanced datasets, as is the case often with clinical data. Determinantal sampling is an important tool not only for Random Forest models, but also for linear regression, where data diversity results in more robust and fair models (*Dereziński et al., 2018*). Moreover, such sampling methods based on DPP are computationally intensive and quantum computers are expected to be useful in this case: quantum computers offer an asymptotic speedup for performing this sampling, and it is expected that next-generation quantum computers will provide a speedup in practice.

We show that, as expected, the quantum version of detDPP-MissForest does not introduce any variance in the downstream classifier when simulated in the absence of hardware noise. While the AUC improvements achieved in our experiments may seem modest, it is the consistency of improvements we observed in our simulation results coupled with removal of variance that makes our approach attractive for clinical data applications where these characteristics are extremely desirable. When implemented on quantum hardware, we observed variance that is caused by the noise in the hardware

itself. More precisely, the output of the quantum circuit is not a sample from the precise determinantal distribution but from a noisy version of it, and this noise depends on the particular quantum circuit implemented and the quality of the hardware. Thus when attempting to compute the highest probability element using samples from the quantum circuit on current hardware, the result is not deterministic. Importantly, unlike for standard MissForest, this variance is not inherent in the algorithm and is expected to reduce considerably with the advent of better quality quantum computers. The quantum circuits needed to efficiently perform determinantal sampling require a number of qubits equal to the batch size used for each decision tree within the Random Forest training and the depth of the quantum circuit is roughly proportional to the number of features. As an example, if we would like to perform the quantum version of the determinantal imputation methods we used for MIMIC-III, then we would need a quantum computer with 150 qubits (the batch size) that can be reliably used to perform a quantum circuit of depth around 400 (the depth is given by $4d \log n$, where $n = 150$ is the batch size and $d = 14$ is the number of features; *Kerenidis and Prakash, 2022*). While quantum hardware with a few hundred qubits that can perform computations of a few hundred steps are not available right now, it seems quite possible that they will be available in the not so far future. In the meantime, further optimization could also help reduce the quantum resources needed for such imputation methods.

While our DPP-based imputation methods can run classically on small datasets such as MIMIC-III, they are computationally intensive and are hard to parallelize due to the sequential nature of the algorithm. This results in less and less efficient imputation for larger datasets where DPP sampling is applied to bigger batches. For example, when a DPP-MICE imputation is run on a dataset of 200 features and batches of size 400, then the training is expected to take multiple hours on a single GPU. The quantum DPP algorithm therefore could provide a way to speed up the hardest part of the imputer using a next-generation quantum computer. For instance, if $d = 200$, and batch size is 400, the number of qubits will be 400 and the depth of the quantum circuit would be $\approx 6400$, whereas it would take $\sim 8 * 10^6$ classical steps for DPP sampling. These are of course simply illustrative calculations and will require more detailed analysis as these machines become available and will need to include parameters such as clock speeds and error correction overheads. Only then can it be experimentally proven that this theoretical asymptotic speedup can translate to a practical speedup for this particular algorithm.

In summary, here we propose novel data imputation methods that, first, improve the widely used iterative imputation methods – MiceRanger and MissForest – as measured by the AUC of a downstream classifier; second, remove the variance of the imputation methods, thus ensuring reproducibility and simpler integration into machine learning workflows; and third, become even more efficient when run on quantum computers. Based on our results, we anticipate an impact of our algorithms on the reliability of models in high-precision value settings, including in pharmaceutical drug trials where they can provide higher confidence in the predictions made by eradicating the stochastic variance due to multiple imputations. In addition, tasks that are currently overwhelmed by the challenges of missingness become more tractable through the approaches introduced here, which is a common problem with real-world-evidence investigations, where detDPP-MICE and detDPP-MissForest can yield improved performance in the face of missingness.

## Methods
### Determinantal point processes (DPPs)
Given a set of items $\mathcal{Y} = \{y_1, \ldots, y_N\}$, a point process $\mathcal{P}$ is a probability distribution over all subsets of the set $\mathcal{Y}$. It is called a determinantal point process (DPP) if, for any subset $Y$ drawn from $\mathcal{Y}$ according to $\mathcal{P}$, we have

$$\mathcal{P}(T \subseteq Y) = \det(K_{T,T}), \tag{1}$$

where $K$ is a real symmetric $N \times N$ matrix, and $K_{T,T}$ is its submatrix whose rows and columns are indexed by $T$. The matrix $K$ is called the marginal kernel of $Y$.

For an $n \times d$ data matrix $A$ and $L = AA^T$, we define the $L$-ensemble $\text{DPP}_L(\mathbf{L})$ as the distribution where the probability of sampling $T$ is

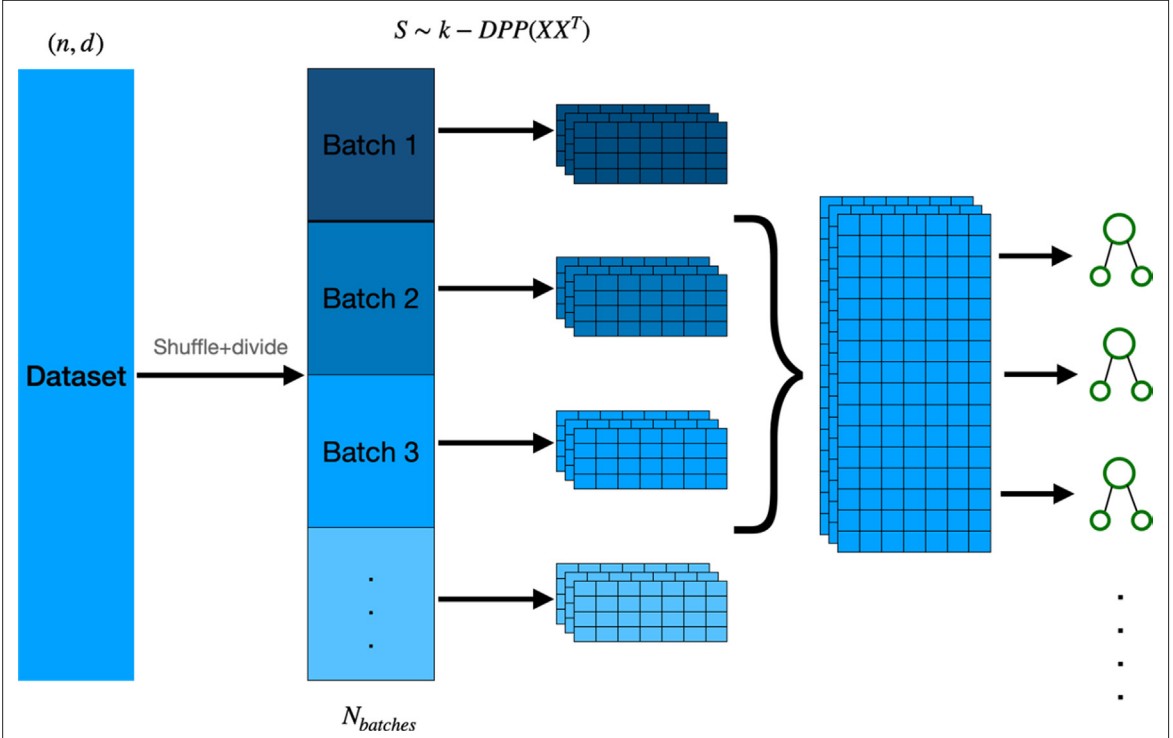

**Figure 4.** The sampling and training procedure for the DPP-Random Forest algorithm: the dataset is divided into batches of similar size, the DPP sampling algorithm is then applied to every batch in parallel, and the subsequent samples are then combined to form larger datasets used to train the decision trees. Since the batches are fixed, DPP sampling can be easily parallelized, either classically or quantumly. DPP, determinantal point processes.

$$P(\{T\}) = \frac{\det(L_{T,T})}{\det(I+L)} \propto \text{Vol}^2(\{a_i : i \in T\}), \tag{2}$$

where $Vol(\{a_i : i \in T\})$ is the volume of the parallelepiped spanned by the rows of $A$ indexed by $T$.

According to this distribution, the probability of sampling points that are similar and thus form a smaller volume is reduced in favor of samples that are more diverse.

An $L$-ensemble is a determinantal point process if $K = L(I+L)^{-1}$.

## Stochastic $k$-DPPs

The distribution $k - \text{DPP}_L(L)$ is defined as an $L$-ensemble which is constrained to subsets of size $|T| = k$.

Different algorithms have been proposed in the literature to sample from $k - \text{DPP}s$, namely *Kulesza and Taskar, 2012*, where sampling $d$ rows from an $N \times d$ matrix takes $O(Nd^2)$ time. There have been improvements over this initial proposal as in *Mahoney et al., 2019*, where there is a preprocessing cost of $O(Nd^2)$ and each DPP sample requires $O(d^3)$ arithmetic operations.

## Deterministic $k$-DPPs

Stochastic DPP sampling may be efficient in practice; however, deterministic algorithms are important for different use cases since they are more interpretable, are less prone to errors, and have no failure probability, which is especially relevant for clinical data (*El Shawi et al., 2019*).

We use a deterministic version of DPP sampling as proposed in *Schreurs and Suykens, 2021* (see Algorithm 1), which is a greedy maximum volume approach. For each deterministic $k - \text{DPP}$ sample, elements with the highest probability are added iteratively. The complexity of the algorithm for selecting deterministically $d$ rows from an $N \times d$ matrix is $O(N^2 d)$ for the preprocessing step and $O(Nd^3)$ for the sampling step.

---

**Algorithm 1 Deterministic $k$-DPP algorithm**

---

**Input:** $N \times N$ Kernel matrix $K \succ 0$, sample size $k$.
**Initialization:** $\mathcal{T} \leftarrow \emptyset$
$V \in \mathbb{R}^{n \times k}$: first $k$ eigenvectors of $K$.
$P = VV^{\mathsf{T}}$

$$p_0(i) = \left\| V^T e_i \right\|^2, \quad i = 1, \dots, k$$

$p \leftarrow p_0$ and $i = 0$
**while:** $i \leq k$ **do**
$\qquad t_i \in \arg\max p$
$\qquad \mathcal{T} \leftarrow \mathcal{T} \cup \{t_i\}$
$\qquad p(j) = p_0(j) - P_{\mathcal{T}j}^T P_{\mathcal{T}\mathcal{T}}^\dagger P_{\mathcal{T}j}, \quad j = 1 \dots n$
$\qquad i \leftarrow i + 1$
**end while**
**Output:** $\mathcal{T}$.

---

## DPP-Random Forest and detDPP-Random Forest

The Random Forest is a widely used ensemble learning model for classification and regression problems. It trains a number of decision trees on different samples from the dataset, and the final prediction of the Random Forest is the average of the decision trees for regression tasks or the class predicted by the most decision trees for classification tasks.

The samples used to train each tree are drawn uniformly with replacement from the original dataset (bootstrapping). The DPP-Random Forest algorithm (see *Figure 4*) replaces the uniform sampling with DPP sampling without replacement.

The running time of the standard Random Forest training on an $N \times d$ matrix is $\tilde{O}(Nd)$, whereas the DPP-Random Forest algorithm takes $\tilde{O}(Nd^2 + d^3)$ steps to run. This shows that while for small $d$ the classical DPP-enhanced algorithms can still be efficient, they quickly become inefficient for larger feature spaces.

Determinantal sampling for regression and classification tasks with full data has been proposed previously for linear regressors (Michał Dereziński and Hsu 2018) and for Random Forest training for a financial data classification use case where it outperformed the standard Random Forest model (*Thakkar et al., 2023*).

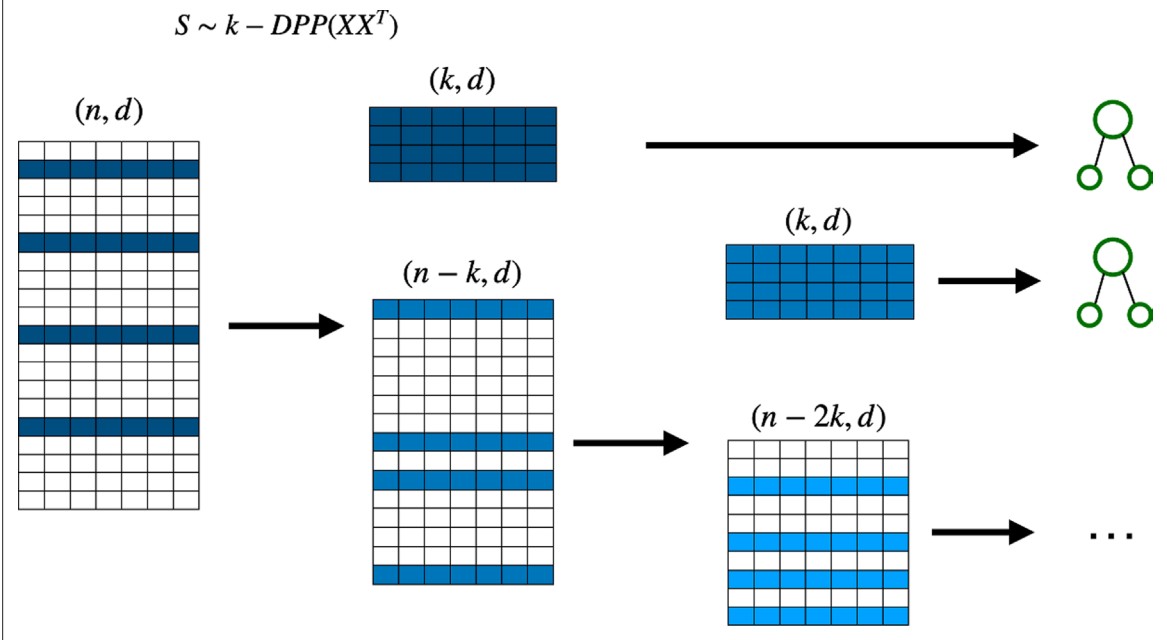

**Figure 5.** Deterministic determinantal point processes (DPP) sampling procedure for training decision trees. At each step, a decision tree is trained using the sample that corresponds to the highest determinantal probability, and which is then removed from the original batch before continuing to the next decision tree.

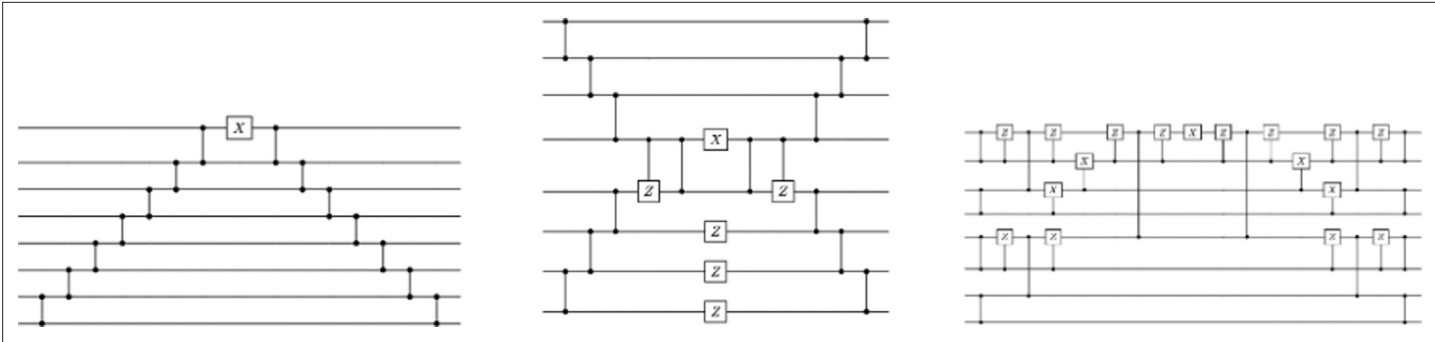

**Figure 6.** Types of data loaders. Each line corresponds to a qubit. Each vertical line connecting two qubits corresponds to a reconfigurable beam splitter (RBS) gate. We also use $X, Z, CZ$ gates. The depth of the first two loaders is linear, and the last one is logarithmic on the number of qubits.

We can also use the deterministic version of DPP sampling for the Random Forest algorithm. This requires removing the sample used at each step (which is the one with the highest probability according to the determinantal distribution) in order to create a smaller dataset from which to sample for the next decision tree (see *Figure 5*). We call this new model detDPP-Random Forest.

Let us note that the distributions of the in-bag DPP samples, which are biased toward diversity, and the out-of-bag (OOB) samples, which reflect the original dataset's distribution, may be different. This could lead to an inaccurate calculation of the OOB error that can be in fact overestimated (*Janitza and Hornung, 2018*). In the DPP-Random Forest case, the batches are stratified and according to the output variable that follows the same distribution as the original dataset. Thus, sampling from different batches could bridge the gap between the in-bag and the OOB distributions. We leave these considerations for future work.

## Quantum methods for DPPs

Quantum machine learning has been a rapidly developing field and many applications have been explored, including with biomedical data, both using quantum algorithms to speedup linear algebraic procedures and through quantum neural networks (*Cerezo et al., 2022*; *Biamonte et al., 2017*; *Landman et al., 2022*; *Cherrat et al., 2022*).

In *Kerenidis and Prakash, 2022*, it was shown that there exist quantum algorithms for performing the determinantal sampling with better computational complexity than the best known classical methods. We describe below the quantum circuits that are needed for performing this quantum algorithm on quantum hardware with different connectivity characteristics and provide a resource analysis for the number of qubits, the number of gates, and the depth of the quantum circuit.

First, we introduce an important component of the quantum DPP circuit, which is the Clifford loader. Given an input state $x \in \mathbb{R}^n$, it performs the following operation:

$$C(x) = \sum_{i=1}^{n} x_i Z^{i-1} X I^{n-i}$$

In other words, it encodes the vector $x$ as a sum of the mutually anti-commuting operators generating the Clifford algebra.

**Table 7.** Summary of the characteristics of the different quantum determinantal point processes (DPP) circuits.
NN = nearest neighbor connectivity.

| Clifford loader | Hardware connectivity | Depth | # of RBS gates |
| --- | --- | --- | --- |
| Diagonal | NN | 2nd | 2nd |
| Semi-diagonal | NN | nd | 2nd |
| Parallel | All-to-all | $4d \log(n)$ | 2nd |

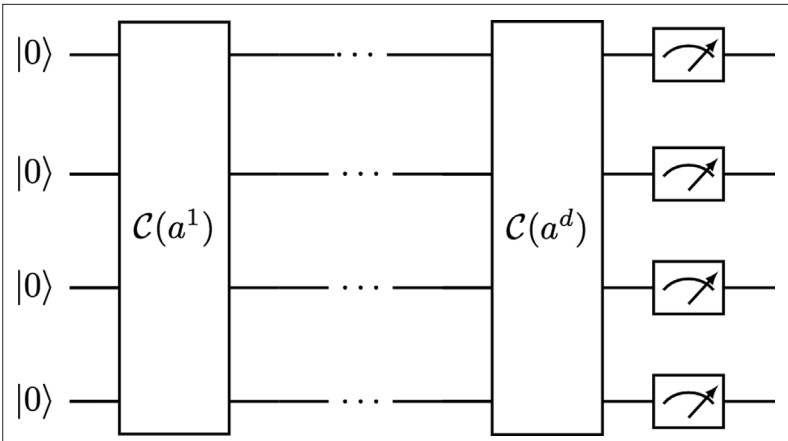

**Figure 7.** Quantum determinant sampling circuit for an orthogonal matrix $A = (a^1, ..., a^d)$. It uses the Clifford loader, which is a unitary quantum operator: $\mathcal{C}(x) = \sum_{i=1}^{n} x_i Z^{i-1} X I^{n-i}$, for $x \in \mathbb{R}^n$.

For implementing this operation with an efficient quantum circuit, we use standard one- and two-qubit gates, such as the *X, Z, CZ* gates as well as a parameterized two-qubit gate called the reconfigurable beam splitter (RBS) gate, which does the following operation:

$$\mathrm{RBS}(\theta) = \begin{pmatrix} 1 & 0 & 0 & 0 \\ 0 & \cos\theta & \sin\theta & 0 \\ 0 & -\sin\theta & \cos\theta & 0 \\ 0 & 0 & 0 & 1 \end{pmatrix} \tag{3}$$

We provide in *Figure 6* three different versions of the Clifford loader that take advantage of the specific connectivity of the quantum hardware, for example, grid connectivity for superconducting qubits or all-to-all connectivity for trapped-ion qubits. These constructions are optimal (up to constant factor) on the number of two-qubit gates. We provide the exact resource analysis in *Table 7*.

We can now use the Clifford loaders described above to perform $k - DPP$ sampling, as described (*Kerenidis and Prakash, 2022*).

Given an orthogonal matrix $A = (a^1, ..., a^d)$, we can apply the qDPP circuit shown in *Figure 7*, which is just a sequential application of $d$ Clifford loaders, one for each column of the matrix, to the $|0^n\rangle$ state, and that leads to the following result:

$$|\mathcal{A}\rangle = \mathcal{C}(a^d) \cdots \mathcal{C}(a^2)\mathcal{C}(a^1)|0^n\rangle = \sum_{|S|=d} \det(A_S)|e_S\rangle$$

Directly measuring at the end of the circuit provides a sample from the correct determinantal distribution.

Both the classical and the quantum algorithms require a preprocessing step with a similar complexity (see *Table 8*), and the improvement using the quantum method achieves a quadratic to cubic speedup

**Table 8.** Complexity comparison of d-DPP sampling algorithms, both classical (*Mahoney et al., 2019*) and quantum (*Kerenidis and Prakash, 2022*).

The problem considered is DPP sampling of $d$ rows from an $n \times d$ matrix, where $n = O(d)$. For the quantum case, we provide both the depth and the size of the circuits.

| | Classical | Quantum |
|---|---|---|
| Preprocessing | $O(d^3)$ | $O(d^3)$ |
| Sampling | $\tilde{O}(d^3)$ | $\tilde{O}(d)$ depth<br>$\tilde{O}(d^2)$ gates |

DPP = determinantal point processes.

in the sampling step. This speedup holds for $n = O(d)$. This is the case for our current implementation of DPP sampling from smaller batches (see *Figure 4*). In addition, the quantum DPP algorithm is efficient in terms of the number of measurements required since one measurement is equivalent to generating one DPP sample.

## Quantum versions of the imputation methods

It is easy to define now a quantum version of the DPP-MICE and DPP-MissForest algorithms, where we use the quantum circuit described above to sample from the corresponding DPP. We can also define a variant of the deterministic algorithms, though here we need to pay attention to the fact that the quantum circuit enables to sample from the determinantal distribution but does not efficiently give us a classical description of the entire distribution. Hence, one can instead sample many times from the quantum circuit and output the most frequent element. This provides a sample with less variance but it only becomes deterministic in the limit of infinite measurements. In the experiments we performed, we used 1000 shots and the samples from the quantum circuits were indeed most of the time the highest probability elements. Of course in the worst case, there exist distributions where, for example, the highest and second highest elements are exponentially close to each other, in which case the quantum algorithm would need an exponential number of samples to output the highest element with high probability. Note though that the quantum imputation algorithm will still have a good performance even with few samples (any high-probability element provides the needed diversity of the inputs), though it will not be deterministic.

## Availability of data and code

The code for the different DPP imputation methods is publicly available at github.com/AstraZeneca/dpp_imp, (copy archived at *AstraZeneca, 2023*). The synthetic dataset can be generated using the *make_classification* method from scikit-learn. The MIMIC-III dataset (*Johnson et al., 2016*) is also publicly available.

## Acknowledgements

This work is a collaboration between QC Ware and AstraZeneca. We acknowledge the use of IBM Quantum services for this work. The views expressed are those of the authors and do not reflect the official policy or position of IBM or the IBM Quantum team.

## Additional information

### Competing interests

Jens Kieckbusch, Philip Teare: are employees of AstraZeneca. The authors declares that no other competing interests exist. The other authors declare that no competing interests exist.

### Funding

No external funding was received for this work.

### Author contributions

Skander Kazdaghli, Conceptualization, Data curation, Software, Visualization, Methodology, Writing – original draft, Writing – review and editing; Iordanis Kerenidis, Conceptualization, Supervision, Methodology, Writing – original draft, Project administration, Writing – review and editing; Jens Kieckbusch, Supervision, Validation, Writing – original draft, Project administration, Writing – review and editing; Philip Teare, Validation, Visualization, Methodology, Writing – original draft, Project administration, Writing – review and editing

### Author ORCIDs

Skander Kazdaghli (iD) http://orcid.org/0009-0005-9044-0919

Reviewer #1 (Public Review): https://doi.org/10.7554/eLife.89947.3.sa1

## Additional files

**Supplementary files**
• MDAR checklist

**Data availability**
The synthetic dataset can be generated using the make classification method from scikit-learn. The MIMIC-III dataset is publicly available at https://mimic.mit.edu/.

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
