## [Editor Report · eLife assessment]

The methods presented in this work provide modest yet consistent accuracy improvements for data classification tasks where certain data are missing. The authors also present a way to use quantum computers for this task. The methodology and results for the classical (non-quantum) case are **solid**, although evidence for the practical quantum advantage via their approach in 'next generation' quantum computers remains **incomplete**. The results are **valuable** and should interest data scientists, life scientists and anyone working in quantum computing.

---

## [Referee Report · Reviewer #1 (Public Review)]

Summary:

The article written by Kazdaghli et al. proposes a modification of imputation methods, to better account and exploit the variability of the data. The aim is to reduce the variability of the imputation results.

The authors propose two methods, one that still includes some imputation variability, but accounts for the distribution of the data points to improve the imputation. The other one proposes a determinantal sampling, that presents no variation in the imputation data, but it seems to be, that they measure the variation in the classification task, instead. As these methods grow easily in computation requirements and time, they also propose an algorithm to run these methods in quantum processors.

Strengths:

The sampling method for imputing missing values that account for the variability of the data seems to be accurate.

Weaknesses:

The authors state "Ultimately, the quality and reliability of imputations can be measured by the performance of a downstream predictor, which is usually the AUC (area under the receiver operating curve) for a classification task." but there is no citation of other scientists doing this. I think the authors could have evaluated the imputations directly, as they mention in the introduction, I understand that the final goal in the task is to have a better classification. In a real situation, they would have data that would be used for training the algorithm, and then new data that needs to be imputed and classified. Is there any difference between imputing all the data together and training the algorithm, versus doing the imputation, training a classifier, then imputing new data (for the testing set), and then testing the classification?

I wonder if there could be some spurious interaction between the imputation and the classification methods, that could bias the data in the sense of having a better classification, but not imputing the real values; in particular when the deterministic DPP is used.